# Bacterial Communities in the Rhizosphere and Phyllosphere of Halophytes and Drought-Tolerant Plants in Mediterranean Ecosystems

**DOI:** 10.3390/microorganisms8111708

**Published:** 2020-10-31

**Authors:** Savvas Genitsaris, Natassa Stefanidou, Kleopatra Leontidou, Theodora Matsi, Katerina Karamanoli, Ifigeneia Mellidou

**Affiliations:** 1Department of Ecology, School of Biology, Aristotle University of Thessaloniki, 54 124 Thessaloniki, Greece; s.genitsaris@ihu.edu.gr; 2Department of Botany, School of Biology, Aristotle University of Thessaloniki, 54 124 Thessaloniki, Greece; natasa.stefanidou@gmail.com; 3Laboratory of Agricultural Chemistry, School of Agriculture, Aristotle University of Thessaloniki, 54 124 Thessaloniki, Greece; cleopatra.leontidou@gmail.com; 4Soil Science Laboratory, School of Agriculture, Aristotle University of Thessaloniki, 54 124 Thessaloniki, Greece; thmatsi@agro.auth.gr; 5Institute of Plant Breeding and Genetic Resources, HAO ELGO-DEMETER, 57 001 Thermi, Greece

**Keywords:** high-throughput sequencing, generalists, rhizobacteria, microbiome, abiotic stress

## Abstract

The aim of the study was to investigate the bacterial community diversity and structure by means of 16S rRNA gene high-throughput amplicon sequencing, in the rhizosphere and phyllosphere of halophytes and drought-tolerant plants in Mediterranean ecosystems with different soil properties. The locations of the sampled plants included alkaline, saline-sodic soils, acidic soils, and the volcanic soils of Santorini Island, differing in soil fertility. Our results showed high bacterial richness overall with Proteobacteria and Actinobacteria dominating in terms of OTUs number and indicated that variable bacterial communities differed depending on the plant’s compartment (rhizosphere and phyllosphere), the soil properties and location of sampling. Furthermore, a shared pool of generalist bacterial taxa was detected independently of sampling location, plant species, or plant compartment. We conclude that the rhizosphere and phyllosphere of native plants in stressed Mediterranean ecosystems consist of common bacterial assemblages contributing to the survival of the plant, while at the same time the discrete soil properties and environmental pressures of each habitat drive the development of a complementary bacterial community with a distinct structure for each plant and location. We suggest that this trade-off between generalist and specialist bacterial community is tailored to benefit the symbiosis with the plant.

## 1. Introduction

The rhizosphere is considered as a microbiota hotspot that hosts a variety of organisms, comprising one of the most complicated and dynamic ecosystems on the planet [1]. Bacterial communities of the roots can be diverse and perform a number of multifaceted functions, equally important for the survival, the growth and the fitness of the plant [2], but also causing suppression of plant development under adverse conditions [3]. Indeed, rhizobacteria produce numerous compounds, which in conjunction with soil properties and environmental conditions, can trigger a dynamic system of interplay at and around the rhizosphere [4]. On the other hand, phyllospheric bacterial assemblages, being among the most ubiquitous microbial communities [5], regulate the processes in the interface between plants and the atmosphere, by controlling the effects of climate, gas composition and the atmosphere’s physical and chemical properties on plants [6].

Soil microbiota provide several ecosystem services and are the lever for facilitating soil processes [7]. Especially soil bacteria are known to be involved in the decomposition of organic matter, nutrient cycling, bioturbation, suppression of soilborne diseases and pests [7], and it is well-documented that they respond to the effects of climate change, such as global warming, increased levels of carbon dioxide, and anthropogenic nitrogen deposition [8]. The bacterial community composition in the rhizosphere of plants in stressed areas, either strongly alkaline—saline soils, or drought areas, can actively exert positive effects on plant survival and growth, and consequently their stress tolerance and alleviation [9]. A large diversity of phylogenetically different bacteria, including the taxa *Pseudomonas*, *Bacillus*, *Enterobacter*, *Streptomyces*, *Klebsiella*, *Agrobacterium*, *Erwinia*, *Azotobacter*, and *Serratia* have been shown to enhance growth and improve the productivity of several crops under salt [10,11,12] and drought stress conditions [13]. The role of these rhizobacteria in plant growth, via several plant-growth-promoting (PGP) traits, such as the production of phytohormones, the supply of atmospheric N, the synthesis of siderophores and of stress-alleviating enzymes, is well-known and documented [14]. However, the potential role of phyllospheric bacteria on plant growth promotion is not yet fully elucidated.

The processes of bacterial colonization differ between the rhizosphere and the phyllosphere [15]. It is well-known that rhizosphere-inhabiting bacteria usually originate from the soil adjacent to the plant root system [16]. Furthermore, the composition, abundance, and structure of the rhizobacterial communities may be influenced by the plant species, the plant’s developmental stage, and the soil properties, such as pH, nutrient availability, and organic content [17]. On the other hand, phyllospheric microbes can arrive and settle on leaves via bioaerosols, raindrop [18], animals [19], and other water or soil resources [20]. Colonizers from other plant parts at early life stages are also reported [21]. Furthermore, evidence indicates that microbe biotic interactions can also shape the bacterial community structure in the different plant compartments, such as the rhizosphere and the phyllosphere [22].

In the past decade, the advances of High-Throughput Sequencing (HTS) approaches have been extensively used to investigate bacterial diversity in various systems [23,24]. These approaches have revealed a vast inter—and intraspecies diversity of 16S rRNA gene sequences and have suggested the existence of highly complex functions and biotic interactions within the microbiomes of plants [25], previously undetected with classical culture techniques [26]. The HTS tools have been used to assess and reveal the whole bacterial diversity of the rhizosphere [27] and the phyllosphere [28] of various plants, indicating that the phyllospheric communities were overall less diverse than the rhizospheric ones [15]. To the best of our knowledge, no studies have been conducted investigating simultaneously the rhizospheric and phyllospheric bacterial communities’ composition and structure of native wild plants grown under naturally stress conditions in saline and drought-stricken areas of the Mediterranean region.

To this end, we collected rhizospheric and phyllospheric samples from native wild plants and from local tomato cultivars grown in three diversely stressed ecosystems of the Mediterranean environment, and investigated their bacterial community diversity and composition by means of 16S rRNA gene high-throughput amplicon sequencing. We examined the rhizospheric and phyllospheric bacterial communities and assessed if they show variability according to sampling location and therefore soil properties, and additionally within location according to each plant separately. Finally, we examined whether the bacterial communities in all plants and sampling sites included consistent and commonly found players and if abundant but specialist taxa that are habitat-specific can be detected.

## 2. Materials and Methods 

### 2.1. Sampling Sites and Sample Collection

Samples were collected from the rhizosphere and phyllosphere of halophytes and drought-tolerant plants, in three Mediterranean areas with different soil properties. The sampling areas were: (i) two different tomato fields in the volcanic Santorini Island (Aegean Sea), namely Vlichada and Emporio; (ii) the peri-urban forest of Seich-Sou located near the metropolitan city of Thessaloniki (Northern Greece); and (iii) the National Park of Delta Axios, also located next to the city of Thessaloniki (Table 1; Appendix A). These sampling sites were considered representative of the adverse ecosystems of the Mediterranean basin (a volcanic island with low precipitation, an ecosystem near an urban area, the Delta of a river), and also containing plant species which dominate such environments, i.e., naturally growing native plants, aromatic and non-aromatic, and cultivated ones, but under xeric conditions, such as the tomatoes of Santorini Island. 

Tomato rhizosphere and phyllosphere samples were collected from eight individuals in each tomato field in the Santorini Island on June 2019, grown under drought conditions. The local tomato cultivar is well-adapted to soils with volcanic properties, high light intensity and temperature, and zero irrigation scheme. No chemical pesticides for common diseases were applied for several weeks prior sampling, while basic fertilization was applied to the soil prior sowing of the seeds. Regarding the other two sampling sites, three characteristic dominant plants from each area, and five individuals from each plant were sampled. In the forest of Seich-Sou, which is characterized by acidic soils, the plants sampled were the drought-tolerant aromatic *Cistus* sp., *Thymus* sp., and *Mentha pulegium* (September 2018), after a period of 38 days with no rainfall in the area (climatological data of HAO DEMETER). In the National Park of Delta Axios, samples from the native halophytes *Sarcoccornia* sp., *Atriplex* sp., and *Crithmum* sp. were collected in June 2018. This wetland covers >33,000 hectares of land, includes estuaries of four rivers and has been included in the Natura 2000 network of European ecological regions [29]. The sampling site in the wetland was located proximate to the estuary of the River Axios, an either saline or sodic environment, characterized by alkaline soils. All plant samples were placed in sterile bags and brought back to the lab under sterile and cold conditions within 6 h. 

### 2.2. Soil Properties

Soil samples were sampled from five different points in each site, and mixed in the same ratio to form a compiled sample. Soil samples were then air-dried, ground to pass a 2-mm sieve and analysed for particle size distribution according to Bouyoucos [30], and for chemical properties according to Sparks [31]. Briefly, pH was measured in a (1:2) soil suspension with water. The electrical conductivity (EC_se_) and the water-soluble cations K, Na, Ca and Mg were determined in the saturation extract. For the analytical determinations of K and Na, flame photometry was used, whereas Ca and Mg were determined by atomic absorption spectrometry. Sodium adsorption ratio (SAR) was calculated from the concentrations of the water-soluble Na, Ca and Mg. All the above methods have been described analytically elsewhere [32]. Furthermore, organic C was determined by the wet oxidation method and CaCO_3_ with a volumetric calcimeter [33]. Soil available NO_3_-N was extracted with 1 M KCl and determined by ultraviolet spectrometry [34]. Olsen-P was extracted with 0.5 M NaHCO_3_, pH 8.5 and determined by the molybdenum blue-ascorbic acid method [35]. Exchangeable K was extracted with 1 M CH_3_COONH_4_, pH 7 and determined by flame photometry [36].

### 2.3. Sample Processing and Sequencing 

To isolate the bacterial communities of the rhizospheric and phyllospheric samples, the bulk soil, and other external material was removed with manual shaking of the root and leaves, and 0.5–1 g from each sample was transferred into phosphate saline buffer (PBS; NaCl 137 nmol L^−1^, KH_2_PO_4_ 1.8 nmol L^−1^, KCl 2.7 nmol L^−1^ and Na_2_HPO_4_ 1.42 nmol L^−1^, pH = 7.4) and sonicated for 10 min (Transsonic 460). The solutions were subsequently centrifuged at 10000 rpm for 20 min, and the sedimentation material was placed in −80 °C until DNA extraction. The rhizosphere samples consisted of both the primary and lateral roots. DNA was extracted from a total of 92 samples (46 rhizospheric and 46 phyllospheric from all plants in all areas), using a NucleoSpin^®^Soil Genomic DNA Isolation Kit (Macherey-Nagel, Bethlehem, PA, USA), according to the manufacturer’s instructions. The concentration and quality of the recovered DNA was checked by a NanoDrop^TM^ spectrophotometer (Thermo Scientific^TM^, Waltham, MA, USA) and confirmed via gel electrophoresis.

The extracted DNA was subjected to PCR using the specific primers targeting the V6-V8 hyper-variable region of the 16S rRNA gene (B969F = ACGCGHNRAACCTTACC and BA1406R = ACGGGCRGTGWGTRCAA) [37]. These primers have been found to successfully amplify approximately 470 bp of all the major high-level bacterial taxonomic groups with up to 83% coverage, after in silico analysis via the SILVA TestPrime 1.0, performed at the Integrated Microbiome Resource (IMR) at Dalhousie University (Halifax, NS, Canada). The PCR products were verified visually by running on a high-throughput Hamilton Nimbus Select robot (Hamilton Company, Reno, NV, USA) using Coastal Genomics Analytical Gels (Hamilton Company) and were cleaned-up and normalized using the high-throughput Charm Biotech Just-a-Plate 96-well Normalization Kit (Charm Biotech, Lawrence, MA, USA). The amplicon samples were sequenced on Illumina MiSeq using 300 + 300 bp paired-end chemistry which allows for overlap and stitching together of paired amplicon reads into one full-length read (http://cgeb-imr.ca/protocols.html). The PCR amplification step and sequencing steps were performed at the Integrated Microbiome Resource (IMR) at Dalhousie University (Halifax, NS, Canada).

### 2.4. Data Analysis

All analyses were performed on the rarefied dataset. Alpha-diversity estimators (the richness estimator S_Chao1_, the Simpson, and Equitability indexes) were calculated for all samples with the PAST 3.16 software [38]. One-way analysis of similarities (ANOSIM) based on Bray-Curtis dissimilarity coefficient was used to examine whether the similarity in bacterial composition of the phyllosphere and rhizosphere between sites was greater than or equal to the similarity within the sites. Then, we used one-way permutational multivariate analysis of variance (PERMANOVA) based on the Bray-Curtis dissimilarity coefficient in the PAST 3.16 software [38] to test whether the soil properties separate the rhizospheric bacterial assemblages of the different sampling sites. To compute beta diversity of the plants’ bacterial assemblages for each sampling site for the rhizosphere and phyllosphere separately, we used the ‘betapart’ R package version 1.5.1 [39]. Beta diversity is a measure of the difference in species composition between two or more local assemblages [40], partitioned into two components according to Baselga et al. [39,41]: (i) spatial turnover in species composition, measured as Simpson dissimilarity (bSIM); and (ii) variation in species composition due to nestedness (bNES) measured as nestedness-resultant fraction of Sorensen dissimilarity. Based on the above, there are two potential ways in which two or more assemblages can be different: one is species replacement (turnover) and the second is species loss or gain, which implies that the poorest assemblage is a subset of the richest one (nestedness). Then, the bacterial assemblages of the different samples (both rhizosphere and phyllosphere samples) were compared using the Plymouth routines in the multivariate ecological research software package PRIMER v.6 [42]. The Jaccard similarity coefficients were calculated based on OTUs presence/absence data, to identify interrelationships between samples and construct an nMDS plot.

Furthermore, OTUs were classified as abundant when their total number of reads exceeded the 0.2% of the total number of reads of the entire dataset (i.e., >900 reads in all samples in these OTUs). In order to identify OTUs with ubiquitous presence in all plants and locations, pointing out to a common bacterial community shared between plants regardless of soil properties, plant characteristics, plant compartment examined, location and environmental conditions, the Levins’ index was calculated [43]. Levins proposed that niche breadth could be estimated by measuring the individuals’ uniformity of distribution among the resource states. For this, specialization of each individual OTU was calculated according to Pandit et al. [44], using Levins’ niche width (*B*) index [43]:(1)$$B=\frac{1}{\sum_{i=1}^{N}P_{ij}^{2}}$$
where *Pij* is the proportion of OTU *j* in sample *i*, and *N* is the total number of samples. Therefore, *B* describes the extent of niche specialization based on the distribution of OTU abundances without considering the abiotic conditions in a local community. The values of the index range between 1 for singletons and a maximum value that varies depending on the dataset, which in our case was 33 (top generalist). The OTUs with *B* index higher than 20 were arbitrarily considered as generalists, while the OTUs with *B* lower than 10 were categorized as specialists [45]. 

### 2.5. Read Processing

The produced reads were subjected to downstream processing using the *mothur* v.1.34.0 software [46], following the proposed standard operating procedure [47]. Briefly, forward and reverse reads were joined, and contigs below 200 bp, with >8 bp homopolymers and ambiguous base calls were removed from further analysis. The remaining reads were dereplicated to the unique sequences and aligned independently against the SILVA 132 database, containing 1,861,569 bacterial SSU rRNA gene sequences [48]. The reads suspected of being chimeras were removed using the UCHIME software [49]. The remaining reads were clustered into Operational Taxonomic Units (OTUs) at 97% similarity level, using the average neighbor method in *mothur*. To obtain a rigorous dataset, OTUs with a single read in the entire dataset were removed from the analysis, as they were suspected of being erroneous sequences [50,51]. The resulting dataset was rarefied with the subsample command in *mothur* v.1.34.0 to 7125 reads, while 11 samples with lower total number of reads were also included in the dataset. We chose this course of action as the best compromise in order to retrieve the majority of the biodiversity detected by sequencing, as rigorous subsampling may result in some OTUs loss, but also to include all samples in the analysis, even those with lower number of reads. Overall, 77 samples (of the 92) were retained for further data analysis, while 15 were excluded because of low sequencing success (Table 1). Taxonomic classification was assigned using BLASTN searches against the SILVA 132 [48], with curated bacterial taxonomy, by applying the lowest accepted level of >80% similarity with a closest relative. The reads belonging to OTUs related to Chloroplasts were removed from the dataset. The raw reads were submitted to GenBank-SRA under the BioProject accession number PRJNA635261.

## 3. Results

### 3.1. Soil Properties in the Sampling Sites

Detailed description of the soil properties in the sampling sites is reported elsewhere [52]. Briefly, the soils collected from the tomato cultivar in the two sampling sites of Santorini (Vlichada and Emporio) were inorganic, coarse in texture, calcareous and alkaline in reaction. They represent ecosystems with different soil fertility based on the contained available NO_3_-N, Olsen P and K: In Emporio, these concentrations were similar or higher than the critical sufficiency range, i.e., 10 mg kg^−1^ NO_3_-N, 10 mg kg^−1^ P and 110 mg kg^−1^ K, respectively, whereas the opposite was evident for Vlichada, in which the concentrations of the three macronutrients were almost five times lower than those of Emporio. Even though they were considered as nutrient-poor and nutrient-rich sites, respectively, they were clustered as more similar compared to the other sampling sites according to Euclidean distances (Figure 1). The soil in the Seich-Sou forest was acidic, enriched with organic matter, and contained adequate amounts of available NO_3_-N and P, but low concentrations of K^+^. On the contrary, the soil from the National Park of Delta Axios was completely different according to Euclidean distances of its properties (Figure 1), and it was an alkaline (pH = 8.1) saline-sodic soil. This was evidenced by the values of Electrical Conductivity (EC) = 69 mS cm^−1^ and SAR 81 [52]. Overall, Vlichada and Emporio were clustered together in terms of Euclidean distances of their soil properties, and the other two locations were dissimilar, with closer to the Santorini sampling sites being the Seich-Sou forest (Figure 1).

### 3.2. Bacterial Communities’ Diversity and Composition

The sequencing pipeline was proven more effective in the rhizospheric samples; it is noteworthy that only two rhizospheric samples were excluded after data processing because of low overall number of sample reads (from *Cistus* sp. and *Thymus* sp. samples in the Seich-Sou forest), while 13 phyllospheric samples (from the Seich-Sou forest and the National Park of Delta Axios) had low sequencing success and thus removed from the dataset (Table 1). For further analysis, 44 out of 46 samples obtained from the rhizosphere (96% of the total rhizosphere samples) and 33 out of the 46 samples from the phyllosphere (72% of the samples) were kept. Rarefaction curves were constructed for all samples, indicating a good coverage of the bacterial communities’ diversity in most of them, and in at least one sample from each plant (Appendix A). The ratio of observed to expected (S_Chao1_) number of OTUs in all samples was >0.55, but in at least one sample per plant the ratio reached >0.75 (Appendix A). A total of 16,355 OTUs were recovered in the entire dataset of 77 samples of the rhizosphere and phyllosphere of the plants sampled.

The sample with the highest OTUs richness was a rhizospheric sample of *Thymus* sp. from the Seich-Sou forest, with 1577 OTUs, and the one with the lowest was a phyllospheric sample of *Crithmum* sp. (60 OTUs) from the National Park of Delta Axios. The average number of OTUs in the rhizosphere and phyllosphere of all the sampled individuals of each plant varied between 65 in the phyllosphere of *Crithmum* sp. (CrPh) and 1209.75 (±268.26 S.D.) in the rhizosphere of *Thymus* sp. (ThRz) (Figure 2, Appendix A), with the majority of plants (both in the rhizosphere and phyllosphere) exhibiting high bacterial diversity (>300 OTUs in average; Figure 2). Relatively low OTUs richness (<300 OTUs in average) was recorded in the phyllospheres of tomato in Vlichada (VlPh), of *Mentha* sp. (MenPh), of *Atriplex* sp. (AtrPh), and of CrPh. The Simpson (1-*D*) dominance index and the Equitability index were high (>0.95 and > 0.70, respectively) in almost all samples except VlPh and CrPh (Figure 2, Appendix A) indicating low dominance of a few OTUs within the bacterial communities, and low variations among the OTUs’ number of reads in each sample.

Overall, 80% of the total number of OTUs recovered from all samples were affiliated to 21 high-level taxonomic groups and the rest were affiliated to unclassified/unidentified Bacteria (data not shown). Among the 21 groups that were detected, eight of them comprised of >70 % of the total number of OTUs (Figure 3). In all samples, Proteobacteria was the dominant group, comprising between 33 and 47% of the total OTUs richness in each plant (either in the rhizosphere or in the phyllosphere), followed by Actinobacteria (8–13% of the total number of OTUs in all samples) and Bacteroidetes (6–11%). Within Proteobacteria, Alphaproteobacteria was the group with the highest OTUs richness (16–22%), followed by Gammaproteobacteria (6–10%) and Deltaproteobacteria (5–7%) (Figure 3). Other less diverse taxonomic groups included the groups Chloroflexi, Deinococcus, Nitrospirae, Deinococcus-Thermus and others with <10 OTUs in the dataset.

### 3.3. Rhizosphere vs. Phyllosphere and Spatial Heterogeneity

Rhizospheric samples were found to be more diverse than phyllospheric ones in all locations and plants, with higher overall OTUs richness in all cases. In addition, commonly found OTUs between the rhizosphere and the phyllosphere of the same plant represented only a portion of the total number of OTUs detected in either plant compartment; overall only 3735 out of the 16,355 were common between the two plant compartments (Figure 4). Evidently, in most of the cases the phyllospheric samples per plant that were included in the analysis were fewer than the rhizospheric ones. In the case of the tomato plants of Vlichada and Emporio and *Thymus* sp. of the Seich-Sou forest (which included the same number of analysed samples for each compartment), the number of OTUs detected in the rhizosphere was much higher (Figure 4). Taking into consideration that the rhizospheric and phyllospheric samples appeared to consist of heterogenous bacterial assemblages between them, the beta diversity of the bacterial communities of all plants in one location was for both compartments higher than 0.85 regardless the sampling site, attributed to turnover of >0.77. The nestedness remained low with the lowest values being computed for the plants in the National Park of Delta Axios (Table 2).

According to one-way ANOSIM the similarity of the bacterial composition within the same site was greater for both the phyllosphere (R  = 0.58, *p*_same_ < 0.001) and the rhizosphere (R =  0.91, *p*_same_ < 0.001) than between the different sites. The significant differences between the site pairs for both the phyllosphere and the rhizosphere was supported by one-way PERMANOVA which detected significant effects of soil properties (F =  4.43, *p*  <  0.001 for the phyllosphere; and F  = 12.9, *p* <  0.001 for the rhizosphere) on the bacterial community structure.

Furthermore, the Jaccard similarity index showed a clear separation of the bacterial assemblages according to the sampling site (Figure 5), thus corroborating PERMANOVA indications on soil properties/location effects on bacterial community structure. Apart from the bacterial assemblages of the tomato plants in Emporio, Santorini, the rhizosphere samples were clearly separated from the phyllosphere samples, further confirming the results of beta diversity estimates and the one-way ANOSIM, that rhizosphere bacterial communities are dissimilar to the respective phyllosphere communities. Finally, smaller groupings of the samples belonging to the same plant/same compartment were also apparent (Figure 5). 

### 3.4. Generalists and Abundant Specialists

According to Levins’ index (*B*), 36 OTUs were found to be generalists (Table 3), either by taking into consideration the entire dataset as a whole, or by calculating *B* in each location, in the rhizosphere and phyllosphere datasets separately and identifying the common generalists in all cases. Euclidean distances based on the number of reads in each generalist OTU in all samples, showed similar groupings as indicated by the Jaccard similarity index (Figure 6), separating sampling sites (thus soil properties), plant species and rhizospheric vs. phyllospheric samples. Most of the generalists belonged to Actinobacteria (12 OTUs out of the 36), followed by Alphaproteobacteria (8/36). Generalist OTUs were also assigned to Bacteroidetes (1 OTU of the 36), Firmicutes (1/36) and Gammaproteobacteria (1/36), while 13 OTUs were assigned to an uncultured bacterial strain (Table 3). 

On the other hand, 22 OTUs were characterized as abundant specialists, meaning they appeared in high abundances in individual samples and thus appeared to be overall abundant with >900 number of reads in the entire dataset, while they were rare or absent in the majority of the samples (Table 4). Among these, 10 belonged to Alphaproteobacteria, six to Gammaproteobacteria, four to Bacteroidetes, one to Firmicutes and one to Actinobacteria. The specialist nature of these abundant OTUs was evident by the habitats of preference, where they increased their number of reads by >20% of the sample’s reads. In the majority of cases they were detected in only one habitat: e.g., OTU_2 and OTU_5/OTU_66, in Vlichada and Emporio site of Santorini Island, respectively; OTU_10, OTU_13, OTU_15, OTU_28, OTU_38, OTU_93, OTU_133, OTU_142 and OTU_144 in the Seih-Sou forest; OTU_7, OTU_9, OTU_22, OTU_37, OTU_61, OTU_68, OTU_103 and OTU_149 in the National Park of Delta Axios. By contrast, only in few cases abundant specialists were detected in certain plant species (e.g., OTU_2, OTU_5 and OTU_66 in tomatoes of Santorini; OTU_10 in *Mentha pulegium*, OTU_15, OTU_28, OTU_142 and OTU_144 in *Cistus sp.*, OTU_5 in *Thymus* sp. of the Seich-Sou forest; OTU_22, OTU_37, OTU_103 in *Sarcocornia* sp., OTU_68 and OTU_149 in *Crithmum* sp. of the National Park of Delta Axios) (Table 4). 

## 4. Discussion

In the present study, the diversity and variability of bacterial communities in the rhizosphere and phyllosphere of native wild plants and local tomato cultivars of three different diversely stressed Mediterranean ecosystems were examined via 16S rRNA gene amplicon sequencing. Overall, in all sampling locations, a large bacterial diversity was revealed. In almost all plant species, the rhizospheric bacterial diversity, in terms of species richness, was much higher than the respective phyllopsheric communities. This is a common finding in similar studies of native and/or cultivated plants in different environments [53,54,55,56,57,58,59,60]. These differences in species richness between the two plant compartments have been attributed to their fundamental physiological and functional differences, as well as the direct impact of their surrounding environments (soil vs air environment). It is well established that the root exudates contribute to the selective growth of specific bacteria [19,56] by signalling plant-microbe and microbe-microbe interactions [61], and can promote the differentiation of the bacterial assemblages through soil-driven selection [59,60]. On the other hand, phyllosphere, being a harsh habitat with shorter lifespan, less nutrient availability [5], and characterized by swift fluctuations in environmental pressures [62], exhibits different physiological functions [18] leading to generally lower bacterial richness and abundance. In the present study, a marked higher richness of bacterial OTUs in the rhizosphere was apparent, reflecting the influence of the stressed habitats and the variability of mechanisms developed by the sampled plants and other co-existing organisms to exploit all available synergistic survival features of the surrounding soil environment. The fact that high-quality DNA was extracted and analysed in more rhizospheric samples, possibly contributing to the higher rhizobacterial OTUs richness, might also be an additional indication of low phyllosphere bacterial abundance.

Our results further indicated that the bacterial communities varied significantly in the three sampling locations. In particular, bacterial communities of the plants in the National Park of Delta Axios, which consists of alkaline and saline-sodic soils, were grouped separately from the samples of the other two locations, according to the Jaccard index, regardless of the plant species and/or the plant compartment. In contrast, bacterial assemblages from the Seich-Sou forest and the Santorini Island exhibited similarities in respect to sampling site (10% similarity, Jaccard index), and stronger on species level within the same site (20% similarity, Jaccard index), probably reflecting similar same soil properties and environmental characteristics. These results are in accordance with the general notion that soil microbial biogeography is primarily controlled by edaphic properties [63], which consequently affect the rhizobacterial communities’ structure in different soils. The bacterial communities of the tomato cultivars of the Santorini Island, on both locations of Vlichada and Emporio, characterized as relatively nutrient-poor and nutrient-rich respectively, showed similarities in the rhizosphere. Local and regional soil properties are identified factors which strongly govern the bacterial community structure of rhizobacteria [64], as they formulate specialized niches favouring the growth of unique bacterial assemblages depending on these properties [63,65]. Indeed, soil properties of the Santorini Island are influenced by the volcanic environment of the island [66], possibly showing convergence and similar soil properties throughout the island, supporting the data of similarities in the rhizospheric communities of the two tomato cultivar fields, independently of the level of soil fertility. However, the bacterial community of the phyllosphere in Vlichada formed a distinct group, suggesting that phyllospheric bacterial assemblages are affected not only by the site, atmospheric conditions and possible air-dispersals [18,67], but also by other factors, such as nutrient availability. As Mello et al. [68] argued, nutrient limitations may have a significant selective pressure on the biodiversity of the microorganisms present in a harsh environment. Indeed, in the nutrient-poor site of Vlichada, shortage of nutrients seemed to affect the phyllospheric communities more than the plant compartment and/or specific location, based on the Jaccard index. These findings need further investigation, taking into consideration that microbe-microbe interactions and within plant compartments microbes’ dispersal are important selective forces forming the microbiomes’ structure of the rhizosphere, phyllosphere and plant endosphere compartments [22].

Although overall the bacterial communities’ composition and structure were similar (in terms of high taxonomic group composition) in all locations, plant species, and plant compartments, dominated mainly by Proteobacteria, followed by Actinobacteria, the 36 generalist OTUs as identified by Levins’ index, belonged mostly to Actinobacteria (>30%). Several studies have shown the dominance of Proteobacteria and Actinobacteria in rhizospheric bacterial communities [69,70]. These groups represent ubiquitous rhizospheric taxa detected in various stressed environments, with many biotechnological applications in sustainable agriculture [71]. Proteobacteria and Actinobacteria were among the top generalists in the phyllospheres of nine perennial plants in a Mediterranean ecosystem [72]. Among the generalist OTUs, taxa closely related to the genera *Bradyrhizobium*, *Steroidobacter*, *Arthrobacter*, *Mycobacterium*, *Pseudonocardia*, *Rhizobium*, *Bosea*, and *Paenibacillus* have been associated with higher nutrient uptake, antioxidant activity [73], catalase activity [74], plant growth [75], phosphate solubilization [76], siderophore production [77], nitrogen fixation [78], cytokinin production [79], and other PGP traits. This consistency points towards a common core of a bacterial community present in different stressed environments of the Mediterranean region, which may establish beneficial interactions with the host plants in a variety and complementary ways.

On the other hand, OTUs that were characterized as abundant, but were found with high number of reads only in few individual samples and were rare or absent from the majority of the samples, were characterized as abundant specialist taxa and comprised of a different taxonomic composition than generalist OTUs. It has been suggested that specialization traits can successfully differentiate bacterial communities [45]. While generalists are able to adapt to varying environmental conditions and thrive in multiple habitats constituting the core of bacterial communities, specialist taxa decrease dramatically in abundance, or even disappear following minor environmental changes [80], thus leading to species sorting, filtered by local environmental conditions [81]. Most abundant specialists of the present study have been detected in high abundances (>20% of the total sample reads) in a single habitat. For example, OTUs related to the taxa *Hymenobacter*, *Methylobacterium*, *Novosphingobium*, and *Phenylobacterium* have been found in high abundances in the drought-stressed system of the Seich-Sou forest. These taxa can also be characterized as indicator taxa, when found in high abundances, for ecosystems with similar soil properties as the Seich-Sou forest [82]. *Hymenobacter* species are UV-resistant bacteria isolated from drought-stressed areas with high UV radiation and low temperatures [83], endosymbiotic *Methylobacterium* mitigates the impact of limited water availability in crops [84], *Novosphingobium* spp. have been associated with dry systems [85], and *Phenylobacterium* spp. have been reported to be favoured by heat [86]. On the other hand, the taxa *Marinobacter*, *Gracilimonas*, *Lewinella* and *Jannaschia* have been detected in >20% of the sample’s reads only in the saline environment of the National Park of Delta Axios, and similarly can be characterized as indicator taxa for similar ecosystems, when in high abundances. These taxa have all been commonly isolated from marine sediments and high-salinity environments [87,88,89,90] and are considered characteristic halotolerant bacteria. Stressed environments overall can play critical roles in structuring the plant bacterial communities by selecting stress tolerant groups of microbes not necessarily correlated directly to the vegetation of the ecosystem [82]. Moreover, it seems that in the studied ecosystems, severity of unfavourable conditions and nutrient availability are both equally important aspects that shape the bacterial communities in terms of richness, structure and behavioural characteristics, which in turn affect plant-bacteria co-existence.

## 5. Conclusions

Knowledge of the composition and structure of the rhizo—and phyllospheric bacterial communities in plants of stressed environments will shed some further light on the richness, variability, dispersal, establishment, functioning of the microbiome on the plant compartments and the specific benefits of this partnership. Our results of 16S rRNA gene amplicon sequencing from rhizo—and phyllospheric samples of native plants in high-saline, dry Mediterranean ecosystems, suggested the presence of overall highly diverse, variable bacterial communities, which differed depending on the plant’s compartment, the soil properties and location of sampling, pointing towards abiotic filtering and environmental drivers. In addition, a commonly found pool of generalist taxa was detected independently of sampling location, plant, or plant compartment. We suggest that these shared OTUs contribute to the plants’ survival and growth through mechanisms for a wide-range and generic management of environmental stresses and complement the variable communities in each soil type which address specific stresses caused because of the plant’s location. The investigation of the driving forces that shape the composition and structure of the rhizo- and phyllospheric bacterial communities in plants under stress will shed further light on the richness, variability, dispersal, establishment, and functioning of the plant microbiome, the role of generalist and specialist taxa, and the specific benefits of plant-bacteria symbioses under harsh conditions.

## Figures and Tables

**Figure 1 microorganisms-08-01708-f001:**
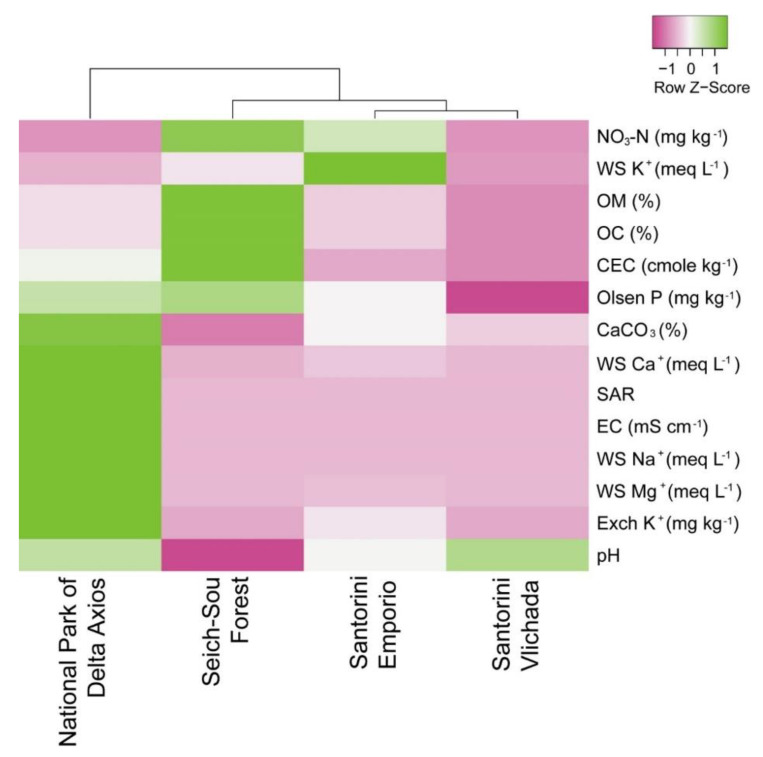
Heatmap of the chemical properties of the soils collected from sampling sites. Columns are mean centred with pink colouring representing low values, and green colouring representing high values. The Average Linage clustering method with Euclidean distances was used to cluster the sampling sites (in respect to soils’ properties). WS, water soluble; OM, Organic Matter; OC, organic C; CEC, Cation Exchange Capacity; SAR, Sodium Absorption Ratio; EC, electrical conductivity of the saturation extract; Exch., exchangeable.

**Figure 2 microorganisms-08-01708-f002:**
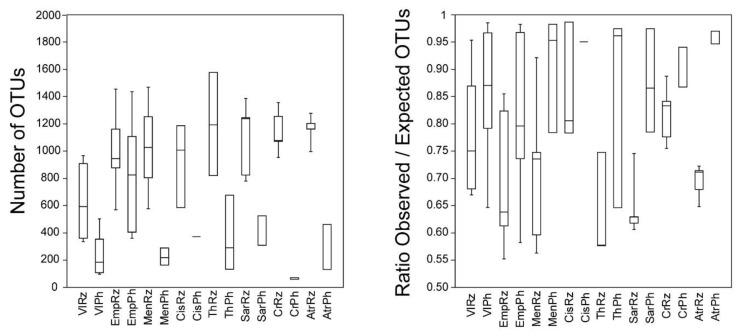
Boxplots of number of OTUs, ratio of observed:expected number of OTUs, the Simpson (1-*D*) and Equitability (H/H_max_) indexes, and the number of reads in the different plant samples. Error bars represent standard deviation of sampled individuals in each plant and plant compartment. See Table 1 for coding assignments.

**Figure 3 microorganisms-08-01708-f003:**
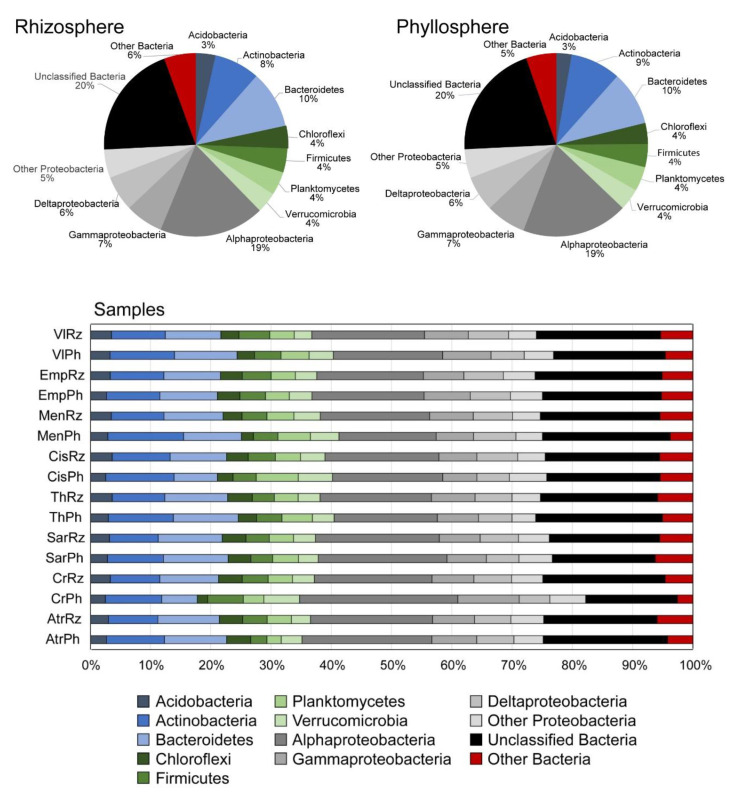
Top charts: Pies of the relative number of OTUs belonging to major high-level bacterial taxonomic groups in the rhizosphere and the phyllosphere of all samples. Bottom chart: Bars of the relative number of OTUs of the major high-level taxonomic groups in each plant regarding separately rhizosphere and phyllosphere samples. See Table 1 for coding assignments.

**Figure 4 microorganisms-08-01708-f004:**
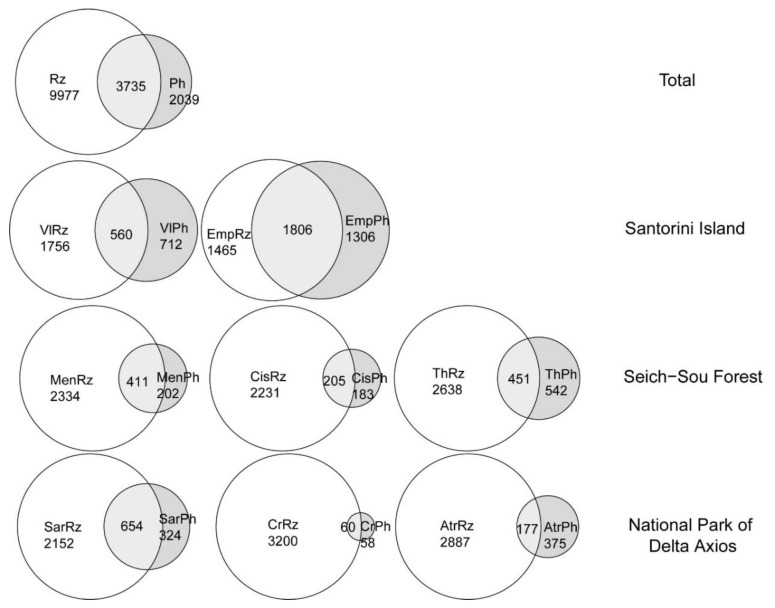
Venn diagrams, illustrating the number of unique and shared OTUs among the total of rhizosphere and phyllosphere samples, and the rhizosphere and phyllosphere samples of each plant. See Table 1 for coding assignments.

**Figure 5 microorganisms-08-01708-f005:**
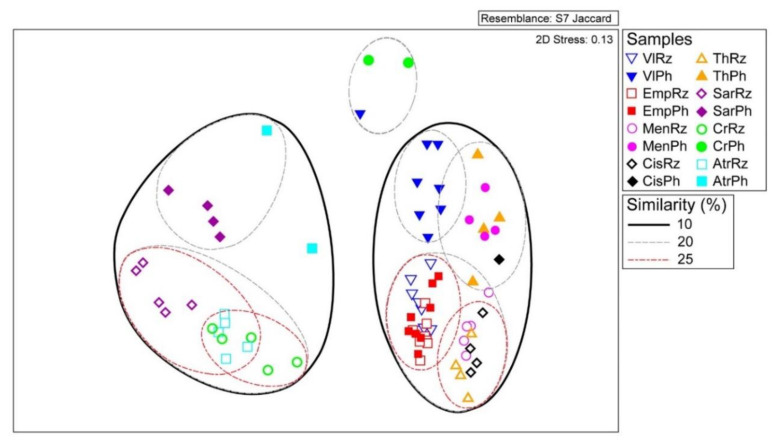
Multidimensional scaling plot of non-transformed OTUs number of reads in all samples (rhizosphere and phyllosphere samples of all individuals sampled), according to Jaccard similarity index. See Table 1 for coding assignments.

**Figure 6 microorganisms-08-01708-f006:**
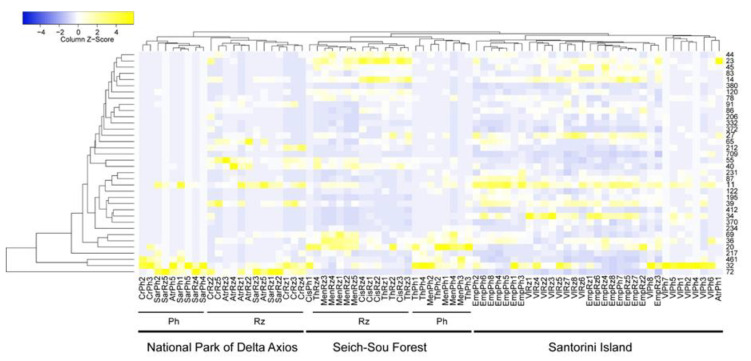
Heatmap of the number of reads of the 36 generalist OTUs as identified by Levins’ (*B*) index in all samples. The Average Linage clustering method with Euclidean distances was used to cluster the sampling sites.

**Table 1 microorganisms-08-01708-t001:** Details on the sampling procedure: Sampling sites, location, plants sampled, part of the plant that samples were collected (phyllosphere or rhizosphere), number of samples left in the dataset after denoizing because of low number of sample reads, and relevant sample code.

Sampling Sites	Latitude (N)	Longitude (E)	Plant Species	Plant Compartment	Number of Samples Sequenced/Analysed after Denoizing	Code
Santorini—Vlichada Santorini—Emporio	36°20′42″	25°26′44″	*Solanum lycopersicum*, ‘Santorini’ (landrace)	Rhizosphere	8/8	VlRz
				Phyllosphere	8/8	VlPh
	36°20′57″	25°25′51″		Rhizosphere	8/8	EmpRz
				Phyllosphere	8/8	EmpPh
Seich-Sou Forest	40°37′41″	22°58′15″	*Mentha pulegium*	Rhizosphere	5/5	MenRz
				Phyllosphere	5/4	MenPh
			*Cistus* sp.	Rhizosphere	5/4	CisRz
				Phyllosphere	5/1	CisPh
			*Thymus* sp.	Rhizosphere	5/4	ThRz
				Phyllosphere	5/4	ThPh
National Park of Delta Axios	40°31′19″	22°39′02″	*Sarcocornia* sp.	Rhizosphere	5/5	SarRz
				Phyllosphere	5/4	SarPh
			*Crithmum* sp.	Rhizosphere	5/5	CrRZ
				Phyllosphere	5/2	CrPh
			*Atriplex* sp.	Rhizosphere	5/5	AtrRz
				Phyllosphere	5/2	AtrPh

**Table 2 microorganisms-08-01708-t002:** Values for beta-diversity (bSOR), turnover (bSIM) and nestedness (bNES) for rhizosphere and phyllosphere bacterial communities of all plants in each sampling site. Bold values indicated the phyllosphere bacterial communities.

	BSOR	BSIM	BNES
Santorini Island	0.87/0.91	0.81/0.82	0.06/0.09
Seich-Sou Forest	0.88/0.85	0.84/0.77	0.04/0.08
National Park of Delta Axios	0.89/0.89	0.87/0.81	0.02/0.08

**Table 3 microorganisms-08-01708-t003:** The OTUs identified as being rhizosphere generalists in the three sampling areas (Santorini Island, Seich-Sou Forest and the National Park of Delta Axios), their putative high-level taxonomic affiliation, their closest relatives based on BLAST searches against the SILVA database and confirmed against NCBI database, and the isolation source of the closest relative.

OTUs	Bacterial Phylum or Class	Closest Relative (% Similarity) [Accession Number]	Isolation Source
OTU_11	*Actinobacteria*	*Blastococcus* sp. (99.5%) [MK239642]	Granite building
OTU_14	*Alphaproteobacteria*	Uncultured Sphingomonadaceae (98.9%) [KC329595]	Ginger cropping soil
OTU_20	*Alphaproteobacteria*	*Sphingomonas yunnanensis* (99.3%) [MN968938]	Culture strain
OTU_23	*Alphaproteobacteria*	*Bradyrhizobium jicamae* (99.3%) [KJ831347]	Culture strain
OTU_27	*Gammaproteobacteria*	*Steroidobacter* sp. (99.1%) [MK311353]	Farmland soil
OTU_32	Unidentified	Uncultured bacterium (99.3%) [KU191639]	Bryophyte
OTU_34	*Actinobacteria*	*Arthrobacter* sp. (99.8%) [MK212372]	Soil
OTU_36	*Actinobacteria*	*Actinoplanes luteus* (99.5%) [NR_145623]	Soil
OTU_39	*Alphaproteobacteria*	*Skermanella aerolata* (98.6%) [MH259920]	*Sargassum horneri*
OTU_40	*Actinobacteria*	*Mycobacterium* sp. (99.1%) [JX273679]	Root
OTU_44	*Actinobacteria*	*Solirubrobacter phytolaccae* (98.6%) [MN686629]	Root
OTU_45	*Actinobacteria*	*Kribbella sandramycini* (99.1%) [MT072122]	Soil
OTU_55	*Actinobacteria*	*Mycobacterium* sp. (99.5%) [KX900598]	Culture strain
OTU_65	*Actinobacteria*	Uncultured *Rubrobacter* sp. (99.5%) [KC110942]	Soil
OTU_69	*Actinobacteria*	*Pseudonocardia* sp. (98.6%) [MN493045]	Root
OTU_72	Unidentified	Uncultured bacterium (99.3%) [JN178597]	Extreme saline-alkaline soil
OTU_78	Unidentified	Uncultured bacterium (98.4%) [JQ978633]	Permafrost soil
OTU_83	*Bacteroidetes*	Uncultured Chitinophagaceae sp. (98.6%) [LN680465]	Coalmine overburden
OTU_86	Unidentified	Uncultured bacterium (98.4%) [MH445072]	Rhizospheric soil
OTU_87	*Actinobacteria*	*Geodermatophilus* sp. (97.9%) [MG200148]	Marine sponges
OTU_91	Unidentified	Uncultured bacterium (99.5%) [MN175141]	Soil
OTU_120	*Alphaproteobacteria*	*Rhizobium* sp. (99.8%) [MT023038]	Culture strain
OTU_122	*Alphaproteobacteria*	*Bosea* sp. (98.2%) [AJ968693]	Culture strain
OTU_195	Unidentified	Uncultured bacterium (97.9%) [EU172577]	Air
OTU_206	Unidentified	Uncultured bacterium (99.1%) [KR560009]	Soil
OTU_212	Unidentified	Uncultured bacterium (98.4%) [KP280904]	Root
OTU_217	*Alphaproteobacteria*	*Neorhizobium alkalisoli* (99.8%) [KF580864]	Root
OTU_231	*Actinobacteria*	*Geodermatophilus aquaeductus* (98.9%) [NR_136840]	Stone
OTU_234	Unidentified	Uncultured bacterium (99.5%) [JF914288]	Seed
OTU_332	Unidentified	Uncultured bacterium (99.1%) [KC331318]	Apple orchard
OTU_370	*Actinobacteria*	*Streptomyces* sp. (99.1%) [MK638452]	Rhizosphere
OTU_372	Unidentified	Uncultured bacterium (98.6%) [JQ049231]	Soil
OTU_380	Unidentified	Uncultured bacterium (99.5%) [MF113653]	Dairy pasteurizer
OTU_412	*Firmicutes*	*Paenibacillus* sp. (99.8%) [KC404044]	Wood core
OTU_461	*Alphaproteobacteria*	*Sphingomonas* sp. (99.3%) [AJ968701]	Culture strain
OTU_709	Unidentified	Uncultured bacterium (98.9%) [AB473917]	Endolithic system

**Table 4 microorganisms-08-01708-t004:** The OTUs identified as being abundant specialists in the entire dataset, their putative high-level taxonomic affiliation, their closest relatives based on BLAST searches against the SILVA database and confirmed against NCBI database, and the isolation source of the closest relative. Shaded boxes denote high relative abundance; i.e >20% of the sample’s total number of reads on average of total sampled individuals in each occasion and different shading colours represent different locations; i.e., black denotes Santorini Island; grey, Seih-Sou forest; light grey, the National Park of Delta Axios.

	Santorini Island	Seich-Sou Forest	National Park of Delta Axios
OTUs	Bacterial Phylum or Class	Closest Relative(% Similarity)[Accession Number]	Isolation Source	VlRz	VlPh	EmpRz	EmpPh	MenRz	MenPh	CisRz	CisPh	ThRz	ThPh	SarRz	SarPh	CrRz	CrPh	AtrRz	AtrPh
OTU_2	*Alphaproteobacteria*	*Sphingobium* sp. (98.9%) [MN181168]	Potato root																
OTU_5	*Gammaproteobacteria*	*Pseudomonas fluorescens* (99%) [CP054128]	Soil																
OTU_7	*Alphaproteobacteria*	Uncultured bacterium (99.6%) [JN038230]	Soil																
OTU_9	*Gammaproteobacteria*	*Marinobacter algicola* (98.6%) [MK493604]	Algal culture																
OTU_10	*Bacteroidetes*	*Hymenobacter* sp. (98.9%) [MH549147]	Plant																
OTU_13	*Alphaproteobacteria*	*Methylobacterium* sp. (99.3%) [MN989088]	Leaves																
OTU_15	*Alphaproteobacteria*	*Novosphingobium lentum* (99.1%) [AB682668]	Culture strain																
OTU_22	*Bacteroidetes*	*Gracilimonas halophila* (98.8%) [NR_158001]	Water																
OTU_28	*Alphaproteobacteria*	*Phenylobacterium* sp. (98.4%) [MF101711]	Hot springs																
OTU_37	*Bacteroidetes*	*Gracilimonas* sp. (99%) [KJ206435]	Saltern																
OTU_38	*Alphaproteobacteria*	Uncultured bacterium (98.6%0 [EU440697]	Soil																
OTU_50	*Gammaproteobacteria*	*Pseudomonas* sp. (99.1%) [CP053697]	Grassland																
OTU_54	*Gammaproteobacteria*	*Lysobacter* sp. (99.3%) [KX230693]	Soil																
OTU_61	*Bacteroidetes*	*Lewinella xylanilytica* (98%)	Culture strain																
OTU_66	*Actinobacteria*	*Streptomyces* sp. (99.1%) [MT538264]	Phosphatic sludges																
OTU_68	*Gammaproteobacteria*	*Pseudoxanthomonas sacheonensis* (99.3%) [MF101054]	Culture strain																
OTU_93	*Alphaproteobacteria*	*Methylobacterium* sp. (99.3%) [MN596044]	Phyllo-sphere																
OTU_103	*Alphaproteobacteria*	*Jannaschia* sp. (98.8%) [FR693293]	Bryozoa																
OTU_133	*Gammaproteobacteria*	*Gilliamella apicola* (98.4%) [MH782109]	Honeybee hindgut																
OTU_142	*Alphaproteobacteria*	*Methylobacterium* sp. (99.3%) [MN989083]	Leaves																
OTU_144	*Alphaproteobacteria*	*Novosphingobium fluoreni* (100%) [KY047400]	Seawater																
OTU_149	*Firmicutes*	Uncultured bacterium (99.3%) [KC110920]	Soil

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
