# Peer review of "Bacterial Communities in the Rhizosphere and Phyllosphere of Halophytes and Drought-Tolerant Plants in Mediterranean Ecosystems"

_microorganisms, 2020, doi:10.3390/microorganisms8111708_

Round 1

Reviewer 1 Report

L48-57 - This fragment seems to me too short. It is worth adding some more information about soil bacteria and why their analysis is important.

L89 - it is necessary to describe how the samples were taken - from what depth, in what amount, how the rhizosphere soil was separated from the rest, whether the samples were pulled or only one from each place where and how they were stored. 

L213-214 - what does it mean "submitted"? That it has not been published? It is not allowed to quote like that. Describe the results.

Figure 2 - You could work on the graphic design of the charts. They are not very encouraging, besides the one with Simpson index is not very readable.

Figure 5 - Where are the signatures of the axles?

Results – I lack a comparison of obtained OTU with soil parameters, correlations to check which parameters determine the diversity of bacteria. I think this should be included in the results and discussed.

FigureS2 – I don't understand this figure. What is it for? What does it bring to paper?

Author Response

We thank the reviewer for his/her positive disposition towards our work and for his/her useful comments and suggestions that helped to improve our original manuscript. All the comments and suggestions have been considered and the necessary changes have been made in the revised text, accordingly. All changes are highlighted in red to facilitate reading. 

L48-57 - This fragment seems to me too short. It is worth adding some more information about soil bacteria and why their analysis is important.

Thank you for your suggestion. We have now added few lines (L48-52) and a couple of references, to add more information on soil bacteria and their involvement in soil processes, highlighting why their analysis is important. All reference numbering has been changed throughout the text accordingly.

L89 - it is necessary to describe how the samples were taken - from what depth, in what amount, how the rhizosphere soil was separated from the rest, whether the samples were pulled or only one from each place where and how they were stored. 

Some further details are included, in accordance with reviewer’s recommendation. The rhizosphere was part of the primary root, with lateral roots included. Sampling isolation procedure is described in 2.3. Sample processing and sequencing. The bulk soil was removed from samples with manual shaking, as indicated in the text. Apart from soil samples which were pooled for soil analyses, rhizosphere and phyllosphere samples were taken from individual plant species as depicted in Table 1 (8 for Santorini samples, and 5 for the rest of sites).

The depth of rhizosphere depended on plant species, as the entire root system was removed, and part of it, consisted of both primary and lateral roots (0.5-1 g), was used for further isolation. In particular, we write in L139-145:

To isolate the bacterial communities of the rhizospheric and phyllospheric samples, the bulk soil, and other external material was removed with manual shaking of the root and leaves, and 0.5-1 g from each sample was transferred into phosphate saline buffer (PBS; NaCl 137 nmol L-1, KH2PO4 1.8 nmol L-1, KCl 2.7 nmol L-1 and Na2HPO4 1.42 nmol L-1, pH = 7.4) and sonicated for 10 min (Transsonic 460)…. The rhizosphere samples consisted of both the primary and lateral roots.

Furthermore, we have added in L121-122: All plant samples were placed in sterile bags and brought back to the lab under sterile and cold conditions within 6 hrs.

And in L124-125 the following: Soil samples from each site were sampled from five different points in each site, and mixed in the same ratio to form a compiled sample.

L213-214 - what does it mean "submitted"? That it has not been published? It is not allowed to quote like that. Describe the results.

We are sorry about this oversight. This paper is actually published, and detailed methods are available. We have added this paper in the reference list, and changed accordingly the reference numbering throughout the text.

Figure 2 - You could work on the graphic design of the charts. They are not very encouraging, besides the one with Simpson index is not very readable.

With respect to the reviewer’ opinion, we are perplexed by this comment. Could you please specify your concerns? We feel that Figure 2 is highly informative highlighting the number of OTUs, the ratio of observed:expected number of OTUs, as well as the Simpson (1-D) and Equitability (H/Hmax) indexes, in the rhizosphere and phyllosphere of all the sampled individuals of each plant, highlighting that the majority of plants (both in the rhizosphere and phyllosphere) exhibited high bacterial diversity. Regarding Simpson index, all data are analytically presented in Supplementary Table S1.

Figure 5 - Where are the signatures of the axles?

Figure 5 represents a Non-metric multidimensional scaling (NMDS) graph, which is an indirect gradient analysis approach producing an ordination based on a distance or dissimilarity matrix. The stress value indicates the number of dimensions (axes) that are needed to describe the similarities between samples (in our case). The stress value being around 0.1 is considered fair to be represented with two axes. Thus, axes basically represent the dimensions of ordination and no axes legends are needed.  

Results – I lack a comparison of obtained OTU with soil parameters, correlations to check which parameters determine the diversity of bacteria. I think this should be included in the results and discussed.

We thank the reviewer for this comment. Actually, we tried to compare OTUs diversity and abundance per sample with environmental variables by running some correlation methods (multiple regression, CCA etc.) but this had no statistically or biological meaning due to the structure of the environmental data (only one measurement of each parameter per site). Thus, we chose to check for significant effects of environmental parameters on overall bacteria community structure and diversity. In particular, we used two-way PERMANOVA to evaluate the accumulated significance of the soil parameters overall on OTUs composition and abundance and two-way ANOSIM to test the significance of these differences among the sites (see L186-192). We acknowledge that in this way we were not able to examine the individual effects of each environmental parameter on bacterial community.

FigureS2 – I don't understand this figure. What is it for? What does it bring to paper?

Figure S2 represents the coverage of diversity with the sequencing effort applied per sample. It is an indication of how much of the bacterial diversity we recovered with the number of sequences retrieved per sample, and it is a commonly used graph in similar studies. We believe that it should be included in our manuscript.

Reviewer 2 Report

The manuscript is prepared very well. Methodology and analysis of results don't raise any objections. I have only some suggestions to improve description of material and methods:

Lines 93 – 95 – what was the reason for choosing these specific locations?

Line 100 - 101 –could you explain why  in the Santorini Island tomato plants were chosen as test plants while in the two other locations they were wild plants? I am also wondering if there weren’t any agricultural practices (such as spraying using pesticides), which could influence bacteria in tomato plants. Maybe you could give some additional informations about these tomato crops?

Author Response

The manuscript is prepared very well. Methodology and analysis of results don't raise any objections. I have only some suggestions to improve description of material and methods:

We would like to thank the reviewer for his/her positive disposition towards our work. All changes, based on the reviewers’ comments, are highlighted in the revised manuscript in red to facilitate reading. 

Lines 93 – 95 – what was the reason for choosing these specific locations?

We feel that these sampling sites are representative of the adverse ecosystems of the Mediterranean basin (volcanic island with low precipitation, the Delta of river), as well as plant species dominating such environments, i.e. naturally growing native plants, aromatic and non-aromatic, and cultivated ones, but under xeric conditions, such as the tomatoes of Santorini. This has been now added in L99-103.

Line 100 - 101 –could you explain why  in the Santorini Island tomato plants were chosen as test plants while in the two other locations they were wild plants? I am also wondering if there weren’t any agricultural practices (such as spraying using pesticides), which could influence bacteria in tomato plants. Maybe you could give some additional informations about these tomato crops?

Essentially, tomato plants grown in Santorini island were selected to represent cultivated annual species grown under xeric conditions, in order to identify differences with the metagenome of wild plants – perennial species, all growing under stressful environments. Ideally, we would have selected the same plant species over the different collection sites, but this was not possible, as the ecosystems under investigation were dominated by different plant species. Thank you for this comment, it is, indeed, helpful to mention any conventional horticultural practices applied in tomatoes. In fact, no chemical pesticides for common diseases were applied for several weeks prior sampling, while basic fertilization was applied to the soil prior sowing of the seeds. The text has been modified accordingly in L110-111.

Reviewer 3 Report

General comments

The manuscript presents the analysis of microbial communities in both the rhizosphere and phyllosphere of halophytes and drought-tolerant plants from Mediterranean. The manuscript was well written and the obtained data was well organized. The key findings of this study were clearly indicated. I suggest this manuscript for publication after revising some minor points.

Specific comment

- The title should be polished. I found something awkward with the phrase "...in Mediterranean ecosystems with different soil properties". It would be clearer if "with different soil properties" is removed. Or "from different soils of Mediterranean".

- Lin 213-214 and line 227: please do not cite the unpublished materials.

- Fig. 1: "Olsen P" should be explained like WS, OM, OC,...

- Please refer to the following “Nomenclature of Microorganisms” to revise throughout the manuscript.

"Names of all bacterial taxa (kingdoms, phyla, classes, orders, families, genera, species, and subspecies) are printed in italics and should be italicized in the manuscript; strain designations and numbers are not."

https://jb.asm.org/content/nomenclature

- Table 3 and Table 4:  the heading "putative high-level taxonomic affiliation" is inappropriate.

The currently listed names are "bacterial class". It can be used as a "heading" of the column.

Author Response

The manuscript presents the analysis of microbial communities in both the rhizosphere and phyllosphere of halophytes and drought-tolerant plants from Mediterranean. The manuscript was well written and the obtained data was well organized. The key findings of this study were clearly indicated. I suggest this manuscript for publication after revising some minor points.

We would like to thank the reviewer for his/her positive disposition towards our work and his/her kind words. All changes, based on the reviewers’ comments, are highlighted in the revised manuscript in red to facilitate reading. 

Specific comment

- The title should be polished. I found something awkward with the phrase "...in Mediterranean ecosystems with different soil properties". It would be clearer if "with different soil properties" is removed. Or "from different soils of Mediterranean".

Thank you for your suggestion. We have now changed the title into: “Bacterial communities in the rhizosphere and phyllosphere of halophytes and drought-tolerant plants in Mediterranean ecosystems’’.

- Lin 213-214 and line 227: please do not cite the unpublished materials.

We are sorry about this oversight. This paper is actually published. We have added this paper in the reference list, and changed accordingly the reference numbering throughout the text.

- Fig. 1: "Olsen P" should be explained like WS, OM, OC,...

Olsen-P refers to the method of extracting phosphorus, and cannot be further explained.

- Please refer to the following “Nomenclature of Microorganisms” to revise throughout the manuscript.

"Names of all bacterial taxa (kingdoms, phyla, classes, orders, families, genera, species, and subspecies) are printed in italics and should be italicized in the manuscript; strain designations and numbers are not."

https://jb.asm.org/content/nomenclature

Thank you for your comment. We have now changed names of bacterial taxa throughout the text according to the “Nomenclature of Microorganisms” as indicated.

- Table 3 and Table 4:  the heading "putative high-level taxonomic affiliation" is inappropriate.

The currently listed names are "bacterial class". It can be used as a "heading" of the column.

Indeed, in the Tables 3 and 4 there are taxonomic affiliations that are either class (e.g. Alphaproteobacterial) or phylum (e.g. Bacteroidetes), thus we changed the heading accordingly.